# HIF-1α Inhibition Improves Anti-Tumor Immunity and Promotes the Efficacy of Stereotactic Ablative Radiotherapy (SABR)

**DOI:** 10.3390/cancers14133273

**Published:** 2022-07-04

**Authors:** Chang W. Song, Hyunkyung Kim, Haeun Cho, Mi-Sook Kim, Sun-Ha Paek, Heon-Joo Park, Robert J. Griffin, Stephanie Terezakis, Lawrence Chinsoo Cho

**Affiliations:** 1Department of Radiation Oncology, University of Minnesota Medical School, Minneapolis, MN 55455, USA; sterezak@umn.edu (S.T.); choxx106@umn.edu (L.C.C.); 2Medical Science Demonstration Center, Korean Institute of Radiological & Medical Sciences, Seoul 01812, Korea; hk0811@kirams.re.kr; 3Department of Radiological & Medico-Oncological Science, University of Science and Technology, Daejeon 34113, Korea; choxx373@umn.edu (H.C.); mskim@kirams.re.kr (M.-S.K.); 4Department of Radiation Oncology, Korea Institute of Radiological & Medical Sciences, Seoul 01812, Korea; 5Department of Neurosurgery, Cancer Research Institute, Hypoxia/Ischemia Disease Institute, Seoul National University, Seoul 03080, Korea; paeksh@snu.ac.kr; 6Department of Microbiology, College of Medicine, Inha University, Inchon 22212, Korea; park001@inha.ac.kr; 7Department of Radiation Oncology, University of Arkansas for Medical Sciences, Little Rock, AR 72205, USA; rjgriffin@uams.edu

**Keywords:** SABR, HIF-1α, HIF-2α, HIF-1α inhibitor, HIF-2α inhibitor, immunotherapy, tumor hypoxia, immune checkpoints

## Abstract

**Simple Summary:**

Stereotactic ablative radiotherapy (SABR), which irradiates tumors with high-dose radiation per fraction, promotes anti-tumor immunity by stimulating various immune processes. SABR also induces vascular damage and obstructs blood flow, thereby increasing tumor hypoxia and upregulation of hypoxia-inducible factors HIF-1α and HIF-2α, master transcription factors for the cellular response to hypoxia. HIF-1α and HIF-2α are key players in the upregulation of immune suppression in hypoxia. Therefore, the radiation-induced increase in anti-tumor immunity is masked by the HIF-mediated immune suppression. Pre-clinical experiments show that inhibition of HIF-1α effectively prevents immune suppression and improves anti-tumor immunity. A combination of HIF-1α inhibitors with immunotherapy with checkpoint blocking antibodies may represent a novel approach to boost anti-tumor immunity and enhance the efficacy of SABR.

**Abstract:**

High-dose hypofractionated radiation such as SABR (stereotactic ablative radiotherapy) evokes an anti-tumor immune response by promoting a series of immune-stimulating processes, including the release of tumor-specific antigens from damaged tumor cells and the final effector phase of immune-mediated lysis of target tumor cells. High-dose hypofractionated radiation also causes vascular damage in tumors, thereby increasing tumor hypoxia and upregulation of hypoxia-inducible factors HIF-1α and HIF-2α, the master transcription factors for the cellular response to hypoxia. HIF-1α and HIF-2α are critical factors in the upregulation of immune suppression and are the master regulators of immune evasion of tumors. Consequently, SABR-induced increase in anti-tumor immunity is counterbalanced by the increase in immune suppression mediated by HIFα. Inhibition of HIF-1α with small molecules such as metformin downregulates immunosuppressive pathways, including the expression of immune checkpoints, and it improves or restores the anti-tumor immunity stimulated by irradiation. Combinations of HIFα inhibitors, particularly HIF-1α inhibitors, with immune checkpoint blocking antibodies may represent a novel approach to boost the overall anti-tumor immune profile in patients and thus enhance outcomes after SABR.

## 1. Introduction

Radiotherapy is a major treatment modality for controlling a variety of cancers. Conventional radiotherapy comprises irradiating tumors with various total doses delivered in 1.5–2.0 Gy/day, 5 days a week. The major radiobiological rationale for treating the cancers with multiple small radiation doses is to take advantage of the greater radiosensitivity of actively dividing cancer cells relative to normal tissues so that tumors are eradicated with acceptable normal tissue damage. In recent years, remarkable technological advances in tumor imaging, radiation delivery, and tumor motion tracking have made it possible to precisely deliver high-dose radiation to tumors with acceptable damage in adjacent normal tissues [1]. During the last decade, an increasing number of cancer patients have been treated with high-dose hypofractionated radiotherapy, which is known as stereotactic radiosurgery (SRS) or stereotactic body radiotherapy (SBRT). Usually, the SRS is referred to the treatment of cranial tumors with 15–25 Gy in a single dose, while SBRT delivers 15–60 Gy in 1–5 fractions to extracranial tumors. SRS and SBRT are collectively called stereotactic ablative radiotherapy (SABR). The clinical outcome of SABR is excellent, with initial local control rates of 80–90% in some tumor types [1]. Unfortunately, however, a tumor may recur, and metastatic growth develops outside the treated area after SABR. Increasing anti-cancer immunity has been suggested to be an approach to improve the ultimate outcome of SABR.

Conventional fractionated radiotherapy with a small dose per fraction sterilizes cancer cells by causing DNA double-strand breaks followed by chromosome aberrations. The cell death mechanisms by SABR have been intensively debated in recent years [2,3,4,5,6,7,8,9,10]. Some investigators have asserted that DNA damage is the major mechanism for the cell death by SABR, while radiobiological principles show that DNA damage alone cannot account for the high efficacy of SABR. The recent accumulating pre-clinical and clinical evidence shows that SABR may eradicate tumor cells through a combination of three pathways: (i) Direct cell death via DNA double-strand breaks, (ii) Indirect cell death via vascular damages, and (iii) Anti-tumor immune response [11,12,13,14,15,16,17,18] (Figure 1). The relative importance of these pathways in eradicating tumor cells and controlling tumors depends on the radiation dose delivered and the tumor type. Cell death due to DNA damage is the major cell death mechanism when the delivered radiation dose per fraction is small, e.g., <5–8 Gy [7,8]. When the radiation dose exceeds 8–10 Gy per fraction, significant indirect cell death secondary to the vascular damage would occur [3,19,20,21,22,23,24,25,26,27,28,29,30,31]. SABR is highly effective because it indirectly kills many radioresistant hypoxic tumor cells by causing vascular occlusion [2,3,4,5,6]. The implication of anti-tumor immune response in the outcome of SABR is unpredictable since the immunogenicity of tumors before irradiation exposure significantly varies depending on tumor type, and high-dose irradiation is a double-edged sword regarding tumor immunity as it can cause both immune stimulation and immune suppression [11,12,13,14,15,16,17,18]. High-dose irradiation of tumors increases the anti-tumor immunity by stimulating practically all known pro-immunologic processes, beginning with the release of tumor-associated antigens (TAAs) from tumor cells and continuing to the final effector phase of killing target tumor cells by activated cytotoxic T-cells [11,12,15,17,18]. However, high-dose irradiation also induces immune suppression by causing vascular damage, increasing tumor hypoxia, and upregulating hypoxia-inducible factor-1α (HIF-1α), the master transcription factor for oxygen homeostasis and immune suppression [3,32,33,34,35]. Hypoxia also upregulates hypoxia-inducible factor-2α (HIF-2α), which has also been known to play an important role in anti-tumor immunity. While the effect of high-dose irradiation on the expression of HIF-1α in tumors has been well-documented, that on the expression of HIF-2α is currently unknown. In this review, we discussed the radiation-induced stimulation of anti-tumor immunity and the radiation-induced suppression of anti-tumor immunity mediated mainly by HIF-1α. A potential approach to overcoming the immune suppression using small-molecule HIF-1α inhibitors is also discussed. 

### 1.1. Vascular Damage in Tumors by SABR

The tumor vascular networks are initially formed through angiogenesis, characterized by the sprouting of arteries and veins of neighboring normal tissues initiated and maintained mainly by vascular endothelial growth factor (VEGF), secreted from the tumor cells and preexisting or convened inflammatory host cells [36,37,38]. Under certain circumstances, circulating endothelial progenitor cells and bone marrow-delivered stem-like cells are assembled to form tumor blood vessels, which is referred to as vasculogenesis [3,39,40]. The endothelial cells of the inner surface of the newly formed tumor blood vessels are frequently disconnected with gaps between the endothelial cells which are occupied by tumor cells and even pericytes [41]. The endothelial layers of tumor blood vessels are supported by incomplete basement membranes and a fragile outer muscle layer. The tumor blood vessels are irregularly constricted or dilated, sharply bent and tortuous [3,19,41]. The supply of essential nutrients including oxygen through such aberrant tumor blood vessels is insufficient for the rapidly growing tumor cell population. Therefore, varying fractions of tumor cells are hypoxic, which are 2–3 times more radioresistant than oxic tumor cells. When tumors are treated with radiotherapy, the tumor blood vessels are also irradiated, and dose-dependent vascular change occurs in the tumor blood vessels. It has been shown that structurally immature tumor microvasculatures are extremely sensitive to ionizing radiation [19]. Irradiation of tumors with doses higher than 8–10 Gy causes dose-dependent vascular damage, resulting in a decline in functional vascularity and blood perfusion [2,19,20,21,22,23,24,25,26,27,28,29,30,31]. For example, in a rat tumor model, functional vascularity declined immediately after irradiation with 2.5 or 5.0 Gy but recovered within 1–2 days, while irradiation with doses greater than 10 Gy caused a severe and long-lasting shutdown of blood perfusion [28]. A recent study using the noninvasive optoacoustic microangiography method demonstrated that 12–18 Gy irradiation of murine tumors in a single dose induced numerous vascular fragmentations and pronounced reductions in the functional vascular density in 1–10 days [30]. Such radiation-induced vascular injuries would inevitably lead to a decline in the supply of oxygen to tumor cells [19,22,30,31], and the resultant hypoxic environment increases the expression of HIFs [2,3,42] (Figure 1).

### 1.2. Effects of High-Dose Irradiation on HIF Expression in Tumors

Hypoxia-inducible factors, HIFs, are heterodimeric transcription factors consisting of labile α-subunits (HIF-1α and HIF-2α) and a stable HIF-1β subunit. HIFs transcribe numerous hypoxia-responsive genes including those involved in energy metabolism, vascularization, invasion, metastasis and immune escape of tumors. HIF-1α and HIF-2α are highly homologous and they share many functional features overlapping, and yet they also exhibit significant differences in tissue-specific expressions and some physiological functions such as the regulation of anti-tumor immune responses [43,44,45,46].

HIF-α is persistently synthesized in the cells but rapidly disintegrates via a pathway initiated by hydroxylation of proline residues by the oxygen-sensitive prolyl hydroxylase domain (PHD) enzyme. This modification leads to ubiquitination of HIF-α and proteasome-mediated degradation in the presence of oxygen [47,48,49]. When HIF-α in cytoplasm moves into the nucleus and binds to nuclear HIF-1β, HIF-α is activated and functions as a master transcription factor of stress responses.

There has been increasing evidence that high-dose irradiation per fraction markedly upregulates the expression of HIF-1α in tumors by decreasing blood perfusion and increasing tumor hypoxia [2,3,42]. For example, irradiation of FSaII fibrosarcoma of C3H mice with 20 Gy in a single dose significantly increased the expression of HIF-1α in 1 day and further increased to almost 4 times that in the control tumors on days 5 after irradiation (Figure 2) [42]. The expression of CA9, a target gene of HIF-1α, and a hypoxia marker, also increased in parallel with that of HIF-1α after irradiation, demonstrating the close association of hypoxia and HIF-1α expression [42]. However, it should be noted that HIF-1α expression can be increased even under a normoxic environment when the proteasomal degradation of HIF-1α is inhibited by physical stresses including reactive oxygen species (ROS) [50,51]. It has been reported that ROS are formed when the hypoxic tumor cells are reoxygenated after irradiation [50]. We have shown that irradiation of FSaII tumors grown in the legs of C3H mice with 10–25 Gy in a single dose induced reoxygenation of the surviving hypoxic cells in 1–5 days [42]. Therefore, the marked increase in HIF-1α expression in the FSaII tumors after irradiation with 15–20 Gy may be attributed to two mechanisms: first, the hypoxic tumor microenvironment (TME) resulting from the severe vascular injuries inhibited the degradation of HIF-1α, and second, ROS generated during the reoxygenation of hypoxic tumor cells prevented the degradation of HIF-1α, leading to a sustained accumulation of functional HIF-1α in TME.

Little is known about the effect of radiation on HIF-2α expression in tumors. Irradiation with 2 Gy increased HIF-2α expression in head-and-neck squamous cell carcinomas (HNSCC) [52]. The expression levels of HIF-2α appeared to be implicated in the resistance of HNSCC to radiotherapy in combination of EGFR inhibition [53]. To the best of our knowledge, there have been no reports on the effect of high-dose-per-fraction irradiation on the expression HIF-2α in tumors. However, it would be reasonable to expect that the expression of HIF-2α in tumors increases, as HIF-1α does, after an exposure to high-dose irradiation due to an increase in hypoxia in TME as a result of the radiation-induced vascular damage.

### 1.3. Stimulation of Antitumor Immunity by High-Dose Irradiation

Conventional radiotherapy, which irradiates tumors 25–50 times with low daily radiation doses for 5–10 weeks, has shown to be immunosuppressive, probably because the immune lymphocytes infiltrated into the tumors are killed by the repeated radiation exposures [11]. The 3D conformal SABR irradiates tumors only 1–5 times in 1–2 weeks, and thus the immune cells in the tumors are less frequently exposed to radiation and thus spared from radiation damage. It has been reported that many cancers are not immunogenic and thus cannot respond to immunotherapy, but the anti-tumor immune responses in these tumors may be stimulated by high-dose irradiation [11,12,13,14,15,16,17,18]. The tumor cell death due to DNA damage and also vascular damage followed by the massive lysis of the dead tumor cells for several days after irradiation with high-dose per fraction may function as a sustained in situ vaccination [3]. Irradiation of tumors increases the cell surface expression of calreticulin and major histocompatibility complex class-1 (MHC class-1), which are crucial for the immune recognition of tumor cells [12,14,15,16]. The heavily irradiated tumor cells release tumor-associate antigens (TAAs), damage-associated Type 1 IFNs, ATP and high-morbidity group box 1 (HMGB1) [16,17,18]. These damage-associated molecular patterns (DAMPs) or “danger signals” are released not only from the tumor cells undergoing immunogenic cell death (ICD) but also from the irradiated tumor-associated host cells. The DAMPs then recruit and activate antigen-presenting cells (APCs), including dendric cells (DCs) by reacting with the associated receptors (Toll-like receptors) on the surface of the APCs. The activated APCs expressing TAAs in the context of MHC migrate to nearby draining lymph nodes and cross-present the TAA to the T-cell receptor (TCR) on the surface of naïve CD8+ T-cells, priming the T-cells to cytotoxic CD8+ T-cells. The processes of acquiring and presenting TAAs to T-cells by DCs require upregulation of HIF-1α in the DCs. HIF-1α has been shown to enhance the synthesis of IFNs in DCs, which promotes the T-cell priming by DCs [54,55]. Upregulation of HIF-1α in DCs was crucial also for the activation of immunosuppressive regulatory T-cells (Tregs) in murine colitis [56]. The CD8+ T-cells are also stimulated (co-stimulated) through the ligation of the CD28 expressed on the surface of the T-cells to the CD80/CD86 on the surface of APCs. The CD8+ T-cells activated in the lymph nodes then migrate to tumors and infiltrate into the tumors. Various cytokines such as IFN-γ, TNF-α, and IL-1β released from the irradiated tumor cells and other immune cells promote the trafficking of the activated T-cells to the tumors. The T-cells infiltrated into the tumors identify the target tumor cells expressing MHC-1 and kill them through two pathways. First, the cytotoxic T-cells injure the membrane of tumor cells by releasing perforin and granzyme, eliciting apoptotic death of tumor cells. Second, the interaction of Fas ligand (FasL) on the CD8+ T-cells with the Fas on tumor cells, which is upregulated by irradiation, activates a caspase cascade, leading to apoptosis of tumor cells [57]. Natural killer cells (NK cells) play a critical role in innate anti-tumor immunity. It has been reported that, whereas low-dose irradiation of tumors enhanced the effects of NK cells, high-dose irradiation impaired NK cell functions [58].

### 1.4. Suppression of Antitumor Immunity by High-Dose Irradiation

#### 1.4.1. Role of HIF-1α in Immunosuppression

It is becoming more and more apparent that the elevated anti-tumor immunity by high-dose irradiation is often counterbalanced or negated by the overwhelming immunosuppressive TME enriched with HIF-1α [11,17,32,33,34,35]. The TME is intrinsically hypoxic due to insufficient blood supply, and it becomes further hypoxic when tumors are exposed to high-dose hypofractionated irradiation and blood vessels are occluded [3,19,30,31]. As addressed before and shown in Figure 2, the increase in hypoxia because of vascular occlusions leads to the accumulation of HIF-1α in the TME [2,3,12,42]. In addition, reoxygenation of fractions of hypoxic cells in tumors after irradiation may also contribute to the increase in HIF-1α expression in tumors after high-dose irradiation [50,51]. Besides promoting tumor cells to adapt to the hypoxic environment and proliferate, HIF-1α also functions as the master regulator of immune escape by transcriptionally upregulating multiple genes that suppress the innate and adaptive immune responses against cancer cells (Figure 2) [16,17,32,33,34,35]. The expression of MHC class-1 is critical for the immune recognition of tumor cells. Whereas irradiation enhances MHC class-1 expression (15), hypoxia and HIF-1α suppress the expression of MHC class-1 [32,59], limiting the tumor cell recognition by APCs. However, hypoxia may increase rather than decrease the expression of MHC class-1 in some cell lines [60]. A hypoxic TME in which HIF-1α is greatly activated stimulates the cancer-associated fibroblasts (CAFs) to release stromal-derived factor-1 (SDF-1), which in turn recruits immature myeloid cells into TME [61]. The recruited myeloid cells then differentiate into tumor-associated macrophages (TAMs) or myeloid-derived suppressor cells (MDSCs). In high-dose irradiated tumors, increasing numbers of immunosuppressive MDSCs were present in avascular hypoxic regions [62], indicating that the irradiation-induced increase in hypoxia and HIF-1α promoted the differentiation of myeloid cells to MDSCs [63]. The transforming growth factor-β (TGF-β) released from the MDSCs stimulates the polarization of TAM into immunosuppressive M2-type macrophages, and also promotes the DC-instructed activation of Tregs [11,12,16,17,32,33,34,35]. Tregs play a major role in immune suppression by secreting immune-suppressive cytokines such as TGF-β, IL-10, and IFN-γ, thereby attenuating the effector T-cell activation and also promoting the immune-suppressive function of MDSCs. The whole process of induction of functional Tregs is closely regulated by HIF-1α [32,33,34,35,64]. NK cell activity is directly influenced by several immune-suppressive cells such as Treg and MDSCs [65]. The NK cells in the irradiated tumor microenvironment are exhausted, wherein the immunosuppressive signals are highly activated by HIF-1α [58]. These results clearly show that HIF-1α plays a critical role in the development of hypoxia-induced immune suppression in tumors treated with SABR.

#### 1.4.2. Role of HIF-2α in Immunosuppression

Contrarily to HIF-1α, relatively little has been revealed on the role of HIF-2α in tumor immune response, although HIF-2α has been known to be implicated in the immunosuppressive functions of TAM. TAMs localize preferentially in hypoxic regions in tumors, where they express HIF-1α and HIF-2α [44]. In a recent study, it was observed that TAMs were polarized to M1 and M2 types by Th1 cytokine (IFN-γ) or Lipopolysaccharide (LPS) and Th2 cytokines (IL-4 and IL-13), respectively, and that mainly HIF-1α was expressed in M1 and mainly HIF-2α was upregulated in M2 macrophages [44]. Functionally, HIF-2α has been shown to directly regulate the expressions of proinflammatory cytokines and chemokine in TAMs and play an essential role in migration of TAMs in tumors [66]. Hypoxic cancer-associated fibroblast (CAFs) in pancreatic ductal adenocarcinoma (PDAC) of mice significantly promoted M2 polarization of TAMs in an HIF-2α-dependent manner in the PDAC, indicating that HIF-2α in CAFs plays a crucial role in hypoxia-related immunosuppression and tumor growth in pancreatic cancer [46]. It is of note that M2-polarized TAMs expressing HIF-2α are a major source of CD86 and PD-L1 (programmed cell death ligand-1), which inactivates cytotoxic CD8+ T-cells. In a recent study, HIF-2α was found to be indispensable for Treg function [67]. Collectively, HIF-2α plays an important role in immunosuppression mainly through controlling TAM-mediated immunosuppression in hypoxic TME. Small-molecule HIF-2α inhibitors such as PT2399, PT2977 and 32–1340 have been shown to poses significant anti-tumor properties and enhance the efficacy of immune checkpoint inhibitors [43,68,69]. Clearly, further investigations are warranted for better understanding of the role of HIF-2α in tumor immune responses.

### 1.5. Immune Checkpoints and HIF-1α

Immune checkpoints protect cancer cells from unregulated attacks by the cytotoxic CD8+ T-cells. The major inhibitory checkpoint molecules are Cytotoxic T-Lymphocyte-associated antigen (CTLA-4), Programed cell death protein 1(PD-1) and its ligand PD-L1 [70]. The expression and activities of these checkpoints are directly and indirectly increased by intratumor hypoxia and HIF-1α [64]. Figure 3A shows that incubation of lung cancer cells in hypoxic environment markedly increased the expressions of HIF-1α and PD-LI, suggesting that the expression of PD-L1 is related to that of HIF-1α. Figure 3B demonstrates that the expressions of HIF-1α and PD-L1 in FSaII tumor cells in vitro increased under chemically induced hypoxia (CoCl_2_), and knockdown of HIF-1α with siRNA markedly reduced the expression of PD-L1. These results unequivocally demonstrate that the expression of PD-L1 is under close control of HIF-1α activity.

The CTLA-4 is expressed on the activated CD8+ T-cells in HIF-1α dependent manner [32,64]. As mentioned before, the co-stimulation of T-cells mediated by the interaction of the CD28 of T-cells with CD80/CD86 on APC is obstructed by the interaction of the CTLA-4 with the CD80/CD86, leading to T-cell exhaustion, anergy and apoptosis [71,72,73]. In preclinical studies, blockade of CTLA-4 with antibodies against CTLA-4 enhanced the anti-tumor immunity by rejuvenating cytotoxic T-cells activity and also enhanced the response of tumors to high-dose hypofractionated irradiation [72,73,74,75]. In clinical settings, immunotherapy using antibodies against CTLA-4 markedly increased the efficacy of SABR for controlling certain types of human tumors [11,13,17,74,75].

PD-1 is highly expressed on the surface of activated CD8+ T-cells [11,17,76,77,78]. VEGF, which is a major downstream target of HIF-1α, has been shown to promote the expression of PD-1 in CD8+ T-cells [79]. TGF-β1, which is upregulated by HIF-1α, has also been reported to increase PD-1 expression on CD8+ T-cells [80,81]. The interaction of PD-1 with its ligand PD-L1 on the surface of tumor cells and other immune cells such as MDSCs, macrophages, DCs induces T-cell exhaustion, anergy and apoptosis. Mounting evidence strongly indicates that the PD-1/PD-L1 pathway plays the most vital role in suppressing immune responses against cancers [77]. Indeed, blocking the PD-1/PD-L1 pathway using antibodies against either PD-1 and PD-L1 alone or combined suppressed the growth of many preclinical experimental tumors and certain types of human tumors and enhanced the outcomes after treatment of tumors with SABR [11,12,13,14,15,16,17,18,32,33,34,35].

### 1.6. Suppression of PD-L1 Expression with HIF-1α Inhibitors

The expression of PD-L1, a transmembrane protein, is highly upregulated in hypoxic regions in tumors where HIF-1α is also highly expressed [82,83]. HIF-1α has been shown to bind to transcriptionally active hypoxia-response elements (HRE) in the PD-L1 proximal promotor and to upregulate the expression of PD-L1 on tumor cells and MDSCs [84]. Given the crucial role of HIF-1α on the expression of immune checkpoints, particularly PD-L1, inhibition of HIF-1α may effectively suppress the PD-1/PD-L1 axis, rescuing the cytotoxic T-cells from exhaustion, anergy and apoptosis. Consistently with such expectations, in a recent study, inhibition of HIF-1α with echinomycin markedly abrogated PD-L1-mediated immune evasion of experimental tumors [85]. Interestingly, inhibition of HIF-1α with echinomycin promoted immune tolerance in normal tissues by increasing PD-L1 via increasing IFN-γ production. Furthermore, the HIF-1α inhibition with echinomycin synergistically increased the immunotherapeutic effects of anti-CTLA-4 antibodies in mouse tumor models [85]. It has also been reported that inhibition of HIF-1α with PX-478, an experimental HIF-1α inhibitor, synergized with anti-PD-1 antibodies to suppress the growth of murine tumors and prolonged the survival of host mice [86]. Taken together, it is highly likely that a combination of HIF-1α inhibitors with immune checkpoint inhibitors would significantly improve anti-tumor immunity [69].

Numerous compounds have been identified to be able to inhibit HIF-1α to date, but none of them have been demonstrated to exclusively target HIF-1α [69,87,88,89,90,91]. We have investigated the ability of metformin and PX-478 to inhibit HIF-1α and suppress PD-L1 expression on tumor cells in recent years. Metformin is one of the biguanides and is widely used to treat type 2 diabetes. Interestingly, it has been found to possess considerable anti-cancer properties and also to significantly increase the response of human tumors to radiotherapy or chemotherapy [92,93]. Such anti-cancer properties of metformin have been attributed to its ability to suppress numerous pathways in energy metabolism. Metformin has also been found to significantly increase anti-tumor immunity by inhibiting HIF-1α activities [94,95]. The mechanism underlying the metformin-induced downregulation of HIF-1α has not been clearly understood, although the process was reported to be oxygen-dependent since metformin suppressed HIF-1α expression in the hypoxic environment but not in the anoxic environment [96]. Importantly, metformin has recently been shown to suppress the expression of PD-L1 [97,98,99]. Because metformin effectively inhibits HIF-1α and that HIF-1α directly controls the expression of PD-L1, it would be rather reasonable to attribute the metformin-induced downregulation of PD-L1 expression to the metformin-induced inhibition of HIF-1α activity. The effects of metformin on the expression of HIF-1α, PD-L1 and PD-1 in FSaII tumors are shown in Figure 4. The expressions of HIF-1α and PD-L1 in FSaII tumors in vivo showed a marked increase 3 days after 20 Gy irradiation, and the daily treatment of host mice with metformin completely abrogated the radiation-induced upregulation of HIF-1α and PD-L1 expression. The expression of PD-1 in the control tumors was considerable, but, unlike the PD-L1 expression, the PD-1 expression decreased 3 days after irradiation, probably because of the death of CD8+ T-cells. However, the number of PD-1-expressing cells in the tumors increased later after irradiation. As shown in Figure 5A, the combined treatment of FSaII tumors with irradiation and daily treatment of host mice with metformin was far more effective than either treatment alone in suppressing the tumor growth. The metformin-induced tumor growth delay and enhancement of the radiation-induced tumor growth delay may be attributable, at least in part, to the suppression of the HIF-1α/PD-L1 pathway and increase in T-cell-mediated anti-tumor immunity.

PX-478 (s-2-amino-3-[4′-N, N, -bis (chloroethyl) amino] phenyl propionic acid N-oxide dihydrochloride) is a recently developed experimental anti-cancer drug that inhibits HIF-1α expression at multiple levels including decreasing mRNA levels and inhibition of HIF-1α translation [100,101,102]. PX-478 has been shown to effectively downregulate both HIF-1α and PD-L1 in experimental glioma cells [82]. As mentioned before, a combination of PX-478 with anti-PD-1 antibody effectively suppressed the growth of murine tumors and prolonged the survival of host mice [86]. Figure 5B shows that PX-478 delayed the growth of FSaII tumors and significantly enhanced the effect of radiation to suppress the tumor growth, which may be attributed, in part, to the suppression of the HIF-1α/PD-L1 pathway.

Taken together, HIF-1α plays a cardinal role in the development of immune suppression in tumors, particularly in the tumors irradiated with high-dose irradiation, and that inhibition of HIF-1α expression with small molecules may be a potentially effective way to prevent immune-suppressive pathways, particularly by preventing the PD-1/PD-L1 pathway (Figure 6). The potential effectiveness of the combination of HIF-1α inhibitors with immune-checkpoint-blocking antibodies to augment anti-tumor immunity in irradiated tumors warrants further investigation.

## 2. Conclusions

SABR markedly increases anti-tumor immunity by promoting a cascade of immune responses, beginning with the release of tumor-associated antigens from tumor cells to the final effector phase of killing target tumor cells by cytotoxic T-cells. Unfortunately, such radiation-induced increases in anti-tumor immunity are counterbalanced by an increase in immunosuppression mediated by HIFα levels (HIF-1α and HIF-2α), which are significantly upregulated in SABR-treated tumors because of increased tumor hypoxia as a result of radiation-induced stress and vascular damage. Therefore, it seems reasonable to expect that effective inhibition of HIFα may enhance the anti-tumor immunity and promote the overall efficacy of SABR in controlling the primary tumor and recurrence and metastatic spread. At present, using antibodies against immune checkpoints PD-1, PD-L1 or CTLA-4 is the main type of immunotherapy available to cancer patients. Unfortunately, however, only a fraction of cancer patients responds to immunotherapy targeting the immune checkpoints. It must be realized that there are other numerous innate and adaptive immune pathways beside the immune checkpoints involved in immune evasion of tumors. In light of the accumulating evidence that HIFα are directly and indirectly involved in many of the innate and adaptive immune pathways including the expression of immune checkpoints, a combination of drugs able to inhibit HIFα, particularly HIF-1α, with antibodies against immune checkpoints may represent a novel and more reliable approach to cancer immunotherapy, especially for patients whose tumors are treated with SABR.

## Figures and Tables

**Figure 1 cancers-14-03273-f001:**
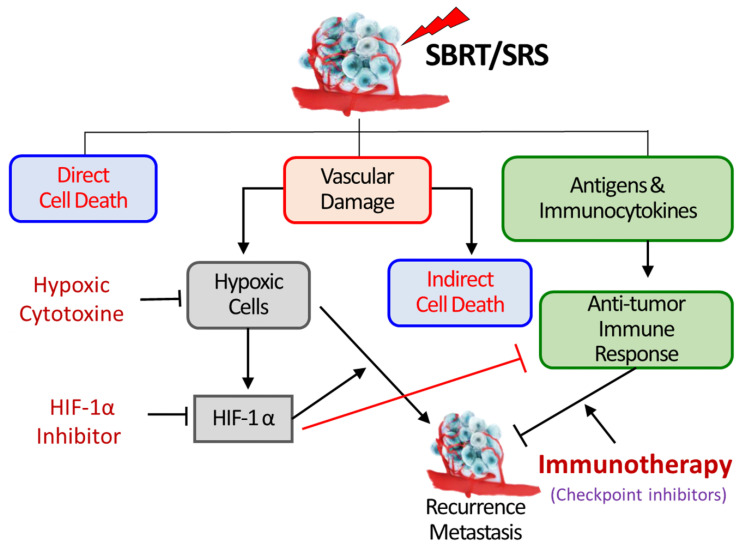
SBRT/SRS (SABR) kills tumor cells by three mechanisms; first—direct DNA damage, second—indirect cell killing via vascular damage, third—promoting immunologic cell killing. The vascular damage increases the expression of HIF-1α in tumor cells and stroma cells, which promotes the proliferation of hypoxic tumor cells leading to recurrence and metastasis. HIF-1α also activates various immune-suppressive pathways, thereby masking the radiation-induced increase in anti-tumor immune response. Inhibition of HIF-1α may suppress the survival and proliferation of hypoxic cells and also stimulate the anti-tumor immune response [3].

**Figure 2 cancers-14-03273-f002:**
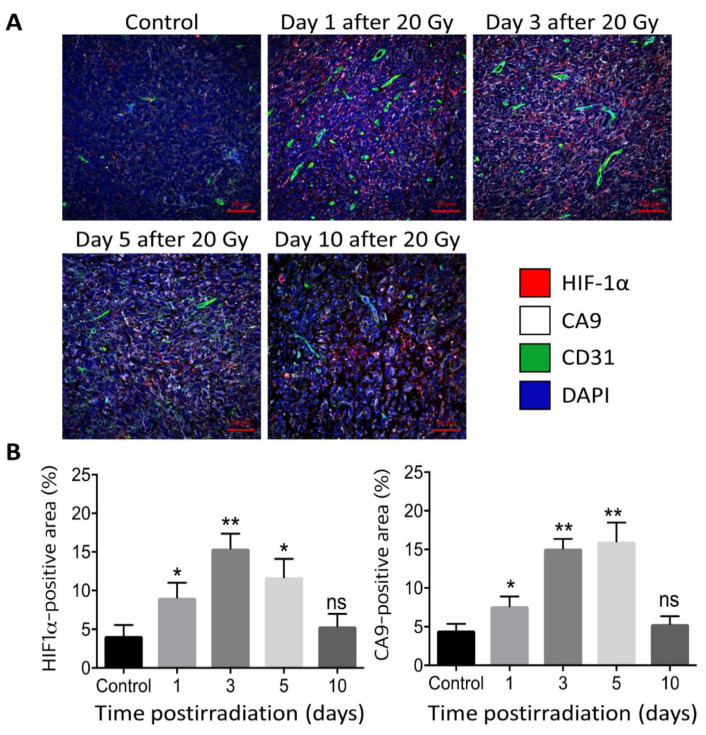
(Upper panel) (**A**). Immunohistochemical image of HIF-1α (magenta), CA9 (white), CD31 (green) and DAPI (blue) in FSaII tumors grown subcutaneously in the legs of C3H mice. (Lower panel) (**B**). % Positive area for HIF-1α and CA9. Irradiation with 20 Gy markedly increased the expression of HIF-1α and its target CA9 in 1–5 days post-irradiation [42]. *, *p* < 0.05; **, *p* < 0.01; ns, not significant, One-way ANOVA.

**Figure 3 cancers-14-03273-f003:**
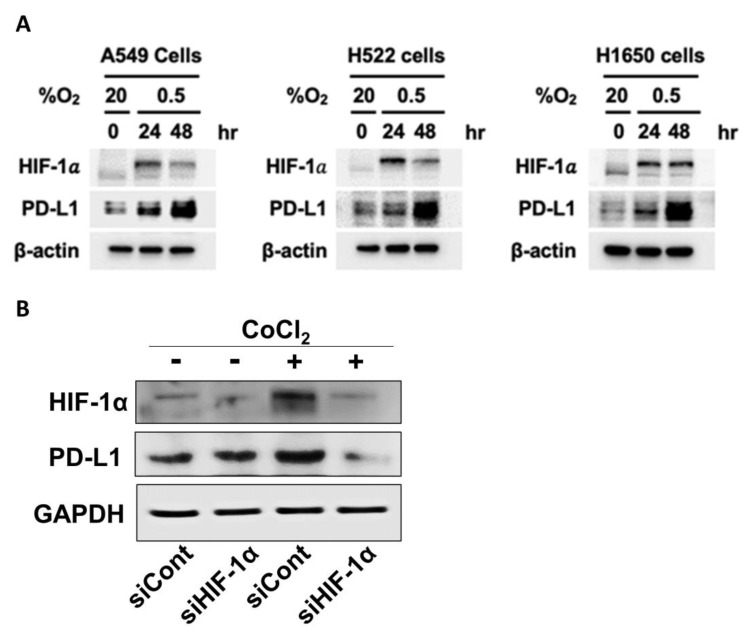
(**A**). Western blot analysis for expression of HIF-1α and PD-L1 in human lung cancer cell lines. A549, H522, and H1650 (obtained from American Type Culture Collection) were incubated under hypoxic conditions (0.5% oxygen) in an InvivO_2_ 500 hypoxia workstation (The Baker Company, Sanford, ME, USA) for 24 and 48 h. Cells were lysed using ice-cold RIPA buffer containing a protease inhibitor cocktail (Roche Applied Science), sodium orthovanadate (Sigma-Aldrich, St. Louis, MO, USA), and sodium fluoride (Sigma-Aldrich). Proteins in whole-cell lysates were resolved by sodium dodecyl sulfate-polyacrylamide gel electrophoresis (SDS-PAGE) and analyzed by immunoblotting. Signals were detected using enhanced chemiluminescence reagents (Pierce). The expressions of both HIF-1α and PD-L1 increased under the hypoxia. (Unpublished results by H. Park). (**B**). Western blot analysis for the expression of HIF-1α and PD-L1 expression in FSaII cells incubated in oxic (DMSO) or hypoxia-mimic medium (250 µM CoCl_2_) for 8 h. Proteins in whole-cell lysates were resolved by sodium dodecyl sulfate-polyacrylamide gel electrophoresis (SDS-PAGE) and analyzed by immunoblotting. Hypoxia increased the expression of HIF-1α and PD-L1 and knocking out of HIF-1α with siRNA downregulated the expression both HIF-1α and PD-L1. (Unpublished results by M. Kim).

**Figure 4 cancers-14-03273-f004:**
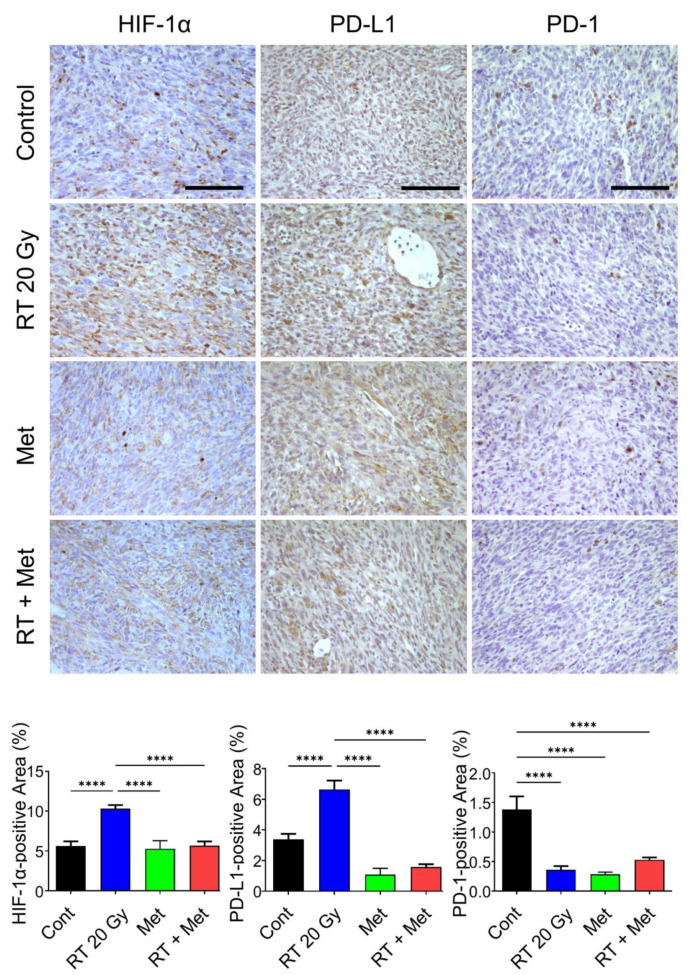
Immunohistochemical protein expression of HIF-1α, PD-L1, and PD-1 in FSaII tumors on day 3 after 20 Gy irradiation. Irradiation markedly increased HIF-1α and daily administration of metformin (150 mg/kg/day) abrogated the radiation-induced increase in HIF-1α. The PD-1 level decreased by irradiation was probably due to death of T-cells expressing PD-1. At least five images per section were captured and the percentage of positive areas was calculated with Image J. The means of tumors ± SD are shown. The scale bars represent 100 μm. ****, *p* < 0.0001, one-way ANOVA.

**Figure 5 cancers-14-03273-f005:**
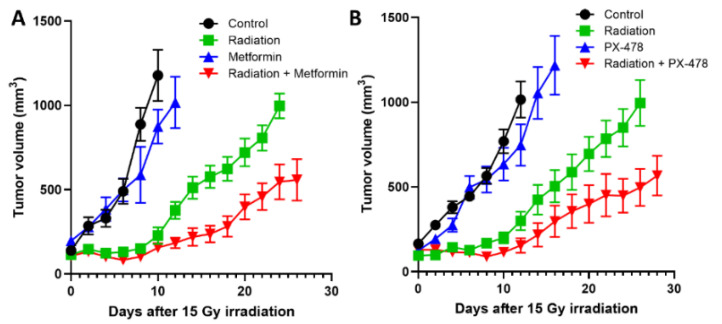
(**A**). Effects of irradiation (15 Gy) and metformin (150 mg/kg/day) alone and combined on the growth of FSaII tumors. (**B**). Effects of irradiation (15 Gy) and PX-478 (35 mg/kg, on days 0, 3, 18, 21) alone and combined on the growth of FSaII tumors. FSaII tumor cells (1 × 10^5^) were subcutaneously injected into the hind legs of C3H mice. When the tumors grew to 0.5–0.7 cm in diameters, they were irradiated with 15 Gy in a single dose using 250 Kv X-rays while the animal were lightly anesthetized with an i.p injection of a mixture of 100 mg/kg ketamine and 10 mg/kg xylazine. Tumor diameters were measured using calipers. The data represent the mean volume of 5–10 tumors ± SE.

**Figure 6 cancers-14-03273-f006:**
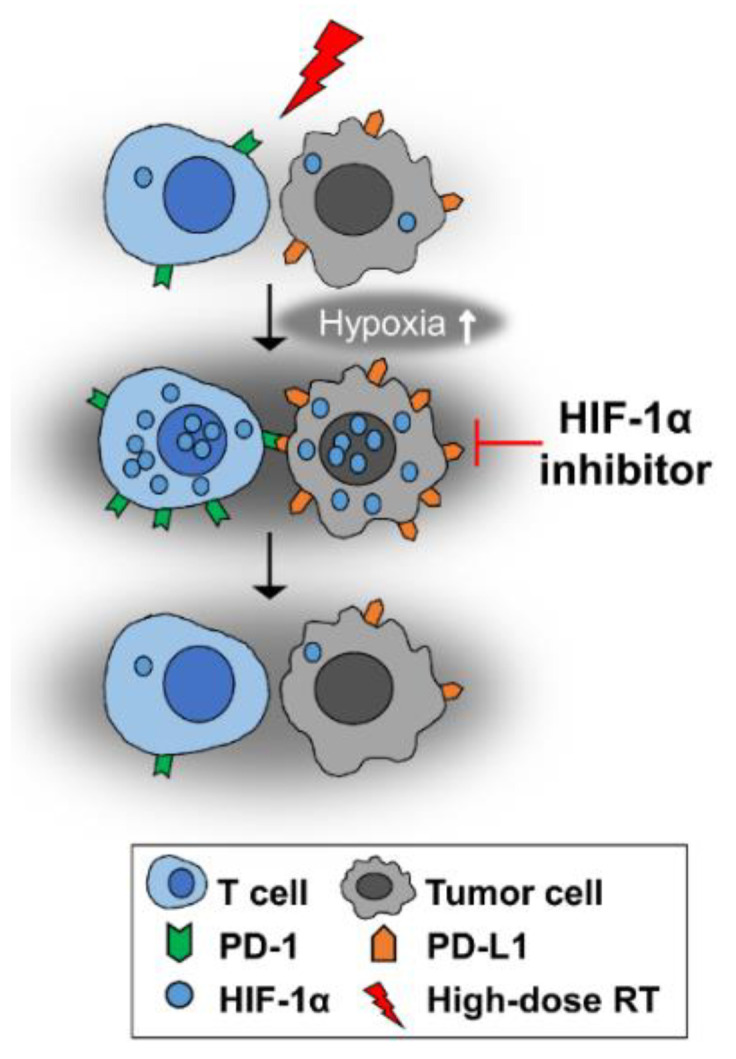
High-dose-per-fraction irradiation causes vascular damage in tumors, thereby increasing hypoxia. Consequently, the expressions of HIF-1α and its downstream targets PD-L1 and PD-1 are increased. HIF-1α inhibition may abrogate the irradiation-induced increases in HIF-1α, PD-L1 and PD-1 expression.

## Data Availability

Data can be requested by contacting the corresponding author.

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
