# Peer review of "HIF-1α Inhibition Improves Anti-Tumor Immunity and Promotes the Efficacy of Stereotactic Ablative Radiotherapy (SABR)"

_cancers, 2022, doi:10.3390/cancers14133273_

Round 1

Reviewer 1 Report

Excellent review about the complex relationship between tumour hypoxia and anti-tumour inmmune response, and the modulation of both phenomena in tumor microenvironment by therapeutic radiation, immune checkpoint inhibitors and HIF-1 inhibitors.

The review concludes in a very interesting hypothesis of radiosensitization and immune potentiation through HIF-1 inhibition.

Extenssive original data (unpublished, non peer-reviewed) have been included. Being pertinent and interesting, it would be better to put together a coherent story with these new pieces of information in a different manuscript. To include the unpublished data in three figures (3, 4 and 5) and to discuss the new results extensively in the text, is a questionable editorial strategy, from my point of view.

Author Response

Please find changes in revised manuscript

Reviewer 2 Report

(Please Use Attached File for Reviewer Commets)

/////////

Unfortunately it is not possible to rank this manuscript adequately as it is not submitted in a complete version. The bibliography is compeletely missing. 

Furthermore, the authors state in l. 86 that the given manuscript is a review article (and the format of the manuscript supports this statement), but it seems to contain original data (although there is no M&M section provided). Figure 2 instead is referenced to another publication [42]. Due to the missing bibliography it is not possible to understand whether this figure has been cited or provided by the authors. The supplementary figure does not exhibit a figure legend. The original images also contain a second supplemental figure which is not provided in the supplemental figure file. There is no reference throughout the manuscript to supplemental data.  

///////

Author Response

Response to “General Remark.

First of all, we thank you very much for the insightful and helpful comments on our manuscript. We tried to comply with all the suggestions as follow.

We were aware of the importance of HIF-2α in anti-tumor immune response. However, since the effect of HIF-2α is relatively less known as compared with that of HIF-1α on anti-tumor immunity, particularly with regards to the effect on checkpoints, we limited our discussed only on HIF-1α in our previous manuscript. However, we totally agree with the comments by the editor, and we thus thoroughly revised the  manuscript discussing the role of HIF-2α in anti-tumor immune response wherever appropriate throughout the revised manuscript.( e.g.,L. 136 – 143; 168 – 177; 261 -284 ).  As discussed in the manuscript, there are virtually no data exist on the effect of radiation on the expression of HIF-2α in contrast to the well-known fact that high-dose irradiation markedly increases the expression of HIF-1α. Therefore, our discussions on the radiation-induced changes in HIF-2α are limited.  

We included some original  unpublished data in this article since we thought that these data are important for our discussion. These data are  only parts of  many results obtained in our ongoing studies on the radiobiological effects of high-dose irradiation on tumor biology. We are preparing several reports on the subjects but we decided to include some of the unpublished observations in this manuscript before they are published as a research paper. As recommended, we included the information on materials and methods for the figures 3 and 5.

To comply with the various recommendation, particularly on HIF-2α,  we added 16 more references in this revised manuscript; References 43-46, 52-56, 60, 66-69, 85,86.

Response to Specific Comments.

L.22, 32

  • The relationship between HIFs and immune suppression is redefined. 
  • The evidence that HIF-2α affects the formation and functions of TAM is  addressed and relevant references are cited ( L 260 – 283: Role of HIF-2α in immunosuppression).
  • Report by Bailey et al. and others relevant articles are addresses. (L. 340– 350)

.

L.82, 175, 181-184.

  • It was an inadvertent mistake. Thank you for your correction.

L.128

  • HIF-1.  It is now changed.

L.135

  • HIF-1a driven transcription. It is now changed.

L.142-143

  • It is compiled with.

L.148, 153

  • Changed to ROS.

L.152

  • TME. Complied with.

L.176-180

  • The sentence regarding DAMPs is divided into two parts.( L. 198 – 204)

  1. 183-185
  • Role of HIF-1a in priming of T-cells. The reports by  Wobben et al. and Fuck et al. on the role of HIF-1α is addressed as references 54, 55. ( L. 206 – 210)

  1. 199
  • Roll. Changed to Role.

L.21,214

  • MHC class-1. The report by Kajiwara et al. is discussed and cited as Ref.60. (L. 240-242)

  1. 221-224
  • TGF-B and DCs. Changed to “ --- and also promotes the DC-instructed activation of Tregs “. ( L. 251).

L.225

  • Immune cytokines. Changed to immunosuppressive cytokines

L.243,244

  • CTLA-4 and VHL knockout cells. In the study by Doedens et al.(previous Ref.54; new reference 64 ),  both the VHL-deficient cells and wild-type cells were used, and  CTLA-4  was expressed on CD8+ cells in the wild-type cells in a HIF-1α dependent manner.  
  • The previous references 56 and 57 are now  references 70 and 71, respectively, in this revised manuscript. As the reviewer pointed out, these articles discussed the role of CTLA-4 , without mentioning HIF-1α.

L.348-353

  • Combination of HIF-1α inhibitors and checkpoint inhibitors. In the recent study by Bailey et al., ( which Editor kindly recommended to include), the potential therapeutic benefit of combining HIF-1α inhibitor with CTLA-4 inhibitors are discussed (Ref .85). Luo et al.(Ref. 86) reported that HIF-1α inhibition and anti-PD-1 antibody synergistically suppressed tumor growth. Based on these reports, we discussed the potential benefit of combining HIF-1α inhibition with checkpoint inhibition in the section “1.6. Suppression of PD-L1 Expression with HIF-1a Inhibition”.

In consistent with such expectation, in a recent study, inhibition of HIF-1α with echinomycin markedly abrogated PD-L1-mediated immune evasion of experimental tumors [85]. Interestingly, inhibition of HIF-1α with echinomycin promoted immune tolerance in normal tissues by increasing  PD-L1 via increasing IFN- production. Furthermore, the HIF-1α inhibition with echinomycin synergistically increased the immunotherapeutic effects of anti-CTLA-4 antibodies in mouse tumor models [85]. It has also been reported that inhibition of HIF-1α with PX-478 , an experimental HIF-1α inhibitor, synergized  with anti-PD-1 antibodies to suppress the growth of murine tumors and prolonged the survival of host mice [86]. Taken together, it is highly likely that a combination of HIF-1α inhibitors with immune checkpoint inhibitors would significantly improve anti-tumor immunity [69].  (L 340-350)

We agree with the reviewer’s comment that  “ ------ this might be because  there are many more HIF-dependent effects that we still do not know yet”. We added following statement at the end of “ Conclusion”.

Unfortunately, however, only a fraction of cancer patients responds to immunotherapy targeting the immune checkpoints. It must be realized that there are other numerous innate and adaptive immune pathways beside the immune checkpoints involved in immune evasion of tumors. In light  of the accumulating evidence that HIFα are directly and indirectly involved in many of the innate and adaptive immune pathways including the expression of immune checkpoints, a combination of drugs able to inhibit HIFα, particularly HIF-1α, with antibodies against immune checkpoints may represent a novel and more reliable approach to cancer immunotherapy, especially for patients whose tumors are treated with SABR. ( L 430-439)

  • Figure 3. In the future study, we will use hypoxic gas instead of CoCl2. We added Figure 3A, which demonstrats that PD-L1 expression increased in HIF-1α dependent manner when human lung tumor cells were incubated in 0.5% O2 atmosphere.

  • Figure 4. The figures for HIF-1α staining are replaced with new ones.

  • The supplementary figure is not needed, and should be deleted.        

Round 2

Reviewer 1 Report

The new version satisfies previous review remarks. 

Reviewer 2 Report

The authors have addressed all my concerns in an acceptable way.